# Paternal family history of premature atherosclerotic disease and perinatal death: A population-based cohort study

**Shwe Sin Win**[1]*, **Kari Klungsøyr**[1,2], **Grace M. Egeland**[1,3], **Gerhard Sulo**[1]

1 Department of Global Public Health and Primary Care, Faculty of Medicine, University of Bergen, Bergen, Norway, 2 Department of Health Promotion, Norwegian Institute of Public Health, Bergen, Norway, 3 Department of Health Registry Research and Development, Norwegian Institute of Public Health, Bergen, Norway

* shwe.win@uib.no

## Abstract

### Background

Studies have reported that pregnancies conceived by fathers with modifiable cardiovascular risk factors are at higher risk of ending in losses compared to those without such risk factors. Our objective was to examine the association between paternal family history _a non-modifiable risk factor_ of premature atherosclerotic disease and perinatal death.

### Methods

This is a population-based cohort study. Information on fathers, aged 18–50 years who participated in three population-based health surveys conducted in Norway during 1974–2003 was linked to their singleton births registered in the Medical Birth Registry of Norway. We used multilevel mixed effect logistic regression analyses with random intercepts by father's identification number. The birth was the observation unit in all analyses.

### Results

A total of 220,386 fathers who had 512,111 births with information on family history of CHD (12.3% with positive family history) and 203,257 births with information on family history of stroke (9.2% with positive family history) were analysed. There were 782 (1.3%) and 195 (1%) perinatal deaths in births to fathers with family history of CHD and stroke while 5,922 (1.3%) and 1,858 (1%) in those without family histories. We found no association between family history of CHD and stillbirth (OR 1.01, 95% CI 0.92; 1.12), neonatal death (OR 0.98, 95%CI 0.86, 1.11) or perinatal death (OR 1.00, 95% CI 0.92, 1.08). Similarly, we found no associations between family history of stroke and stillbirth (OR 1.00, 95% CI 0.82, 1.21), neonatal death (OR 1.09, 95%CI 0.84, 1.41) or perinatal death (OR 1.02, 95% CI 0.88, 1.20).

### Limitations

Socioeconomic background of fathers was captured using imperfect proxy. Potential misclassification of family history and selection bias should be considered.

**Data Availability Statement:** Although the authors have no access to directly identifiable information, the data used in the present study are defined as

potentially identifiable, and they include sensitive information (perinatal loss and health issues). They are based on family data – generational data where the father is linked to his own children and this data structure is not possible to make anonymous or even de-identified. In Norway, there are both ethical and legal restrictions on sharing potentially identifiable dataset: ACT 2008-06-20 no. 44: Act on medical and health research (the Health Research Act) and GDPR- General Data Protection Regulation. However, the data are available for other researchers upon request to the Norwegian Institute of Public Health (https://helsedata.no/) under the approval of the Regional Ethic Committee (https://rekportalen.no/#hjem/home).

**Funding:** This study is funded by non-commercial, non-government organizations Stifelsen Dam and landsforeningen uventet barnedød under project number 2022/FO387075. GE and KK received funding. The funders played no role in research.

**Competing interests:** NO authors have competing interests

## Conclusion

Results of this large, cohort study including half-a million births, do not indicate an association between paternal family history of premature atherosclerotic disease and perinatal death.

## Introduction

Until 1975, the World Health Organization (WHO) defined the stillbirth rate as the number of babies born with no signs of life weighing 1000 grams or more or being born at 28 gestational weeks or more per 1000 total births [1]. After that, stillbirth is defined as the expulsion of a foetus with no signs of life weighing 500 grams or more at 22 weeks or more of gestation. The perinatal mortally rate is usually defined as the combination of stillbirths and early neonatal deaths during the first week of life per 1000 births (stillbirths and livebirths) [2]. The perinatal mortality rate is an important indicator often used for monitoring the quality of prenatal, intra partum and new-born care [3].

Using 1000 grams as cut-off for stillbirth, there were 6.6 perinatal deaths per 1000 births in Europe in 2017 [4]. In Norway, the rate was 2.4 per 1000; slightly lower than that of all Nordic countries' 2.6 per 1000. Although perinatal mortality rate is low in the Nordic countries, losing a child is a devastating experience for the parents, and there is a need for preventing these outcomes.

The causes of stillbirth are multifactorial, and include foetal (e.g., birth defects, syndromes, and aberrant growth), placental (e.g. chorioamnionitis, umbilical cord accidents, abruption, infarction), and maternal (e.g. obesity, hypertension, diabetes mellitus, autoimmune disorders, intrahepatic cholestasis of pregnancy) causes [5]. Still, in up to 30% of cases, the causes remain unexplained even with complete clinical and autopsy records [6, 7]. The causes of neonatal death include factors such as prematurity, multifetal pregnancies, birth defects, and intrapartum-related complications such as birth asphyxia and infections [5, 8].

Most studies on parental causal factors for stillbirth and perinatal death have focused maternal and pregnancy related factors, however some have also explored the role of fathers. Studies have reported that paternal age [9–11], education [12], occupation [6], lifestyle (diet [9, 13], smoking [13–17], drinking [13]), exposure to drugs [18] and toxins [9, 19] in the environment may be related to miscarriage [9, 10, 13–15, 17], stillbirths [6, 11–13, 19], and perinatal death [18].

Further, paternal factors are also found to be associated with several pregnancy complications known to increase the risk of perinatal loss: Studies have reported paternal effects on placental vascular dysfunction related to pregnancy complications such as pre-eclampsia, and fetal growth restriction [20–26]. Moreover, there is familial aggregation of these complications. For example, both women and men born in pregnancies complicated by pre-eclampsia had higher risks of having pre-eclamptic pregnancies themselves [22]. These findings suggest that genes responsible for placental vascular dysfunction and certain placental characteristics can also be passed from the father to the fetus [22, 26, 27].

Recently, a study reported that the risk of pregnancy losses (i.e., ectopic pregnancy, spontaneous abortion or stillbirth) increased with increasing numbers of cardiovascular risk factors in the fathers [28]. Such increase in risk could either be due to direct effects by these factors on the semen, but could also be due to underlying common genetic factors of importance for both cardiovascular factors and perinatal loss. While the focus in studies on paternal risk

factors for perinatal loss so far has been on paternal modifiable risk factors, evidence on the potential association between paternal non-modifiable factors and pregnancy loss is still lacking.

Family history of premature coronary heart disease (CHD)_ a non-modifiable risk factor- reflects genetic susceptibilities, shared environment, and common behaviors [29]. Genetic susceptibilities could include factors such as hemostatic dysfunction that are assumed to have shared underlying genetic or epigenetic aetiologic mechanisms as pregnancy loss or complications during pregnancy and CHD [30–35]. A few studies examined associations between maternal family history of atherosclerotic diseases and pregnancy loss with inconsistent results [31, 32, 36, 37]. However, no studies have examined these associations in fathers even though fathers contribute 50% of genome to an embryo and there are well known effects of paternal genes (as expressed in fetus) on placental vascular dysfunction [20–26].

To sum up, previous studies have documented associations between several sociodemographic, lifestyle and environmental factors in fathers and the risk of perinatal loss. Further, paternal factors have been found to influence the risk of pregnancy complications associated with perinatal loss. The risk of pregnancy loss increased with increasing number of cardiovascular risk factors in fathers, this could be explained by common underlying aetiologic factors. Family history of cardiovascular disease reflects genetic susceptibilities, shared environment, and common behaviors and may be used to highlight potential common underlying aetiologic mechanisms for both cardiovascular risk and perinatal loss. So far, no studies have focused paternal family history of cardiovascular disease and the risk of perinatal loss. This study, therefore, aimed at exploring the potential association between paternal family history of premature coronary heart disease or stroke and stillbirth, neonatal death, and perinatal death that may share common underlying aetiologic mechanisms.

## Materials and methods

### Data sources

We used data collected from three partially overlapping population-based health surveys, spanning over 30 years (1974–2003), namely the Screening of 40-year-old program ('HU40', 1985–1999) [38], the Three County Health Surveys of 35-49 years old individuals ('HU3', 1974–1988) [39], and Cohort of Norway (CONOR, 1994–2003) [40].

### Screening of 40-year-old program

The HU40 Program was a nationwide cardiovascular health survey conducted by Norwegian health authorities in the 1980s and 1990s, including over 400, 000 Norwegian men and women aged 40–43 years. Participation rates ranged from 52% to 88%, depending on year and county.

### Three county health survey

HU3 was a cardiovascular disease screening program for 35–49 years old Norwegians conducted in the counties of Finmark, Sogn og Fjordane and Oppland between 1974 and 1988 with a participation rate of 95.4%.

### Cohort of Norway

CONOR is a national database that merged regional data from 10 epidemiological studies using questionnaires and a short health examination conducted between 1994 and 2003 and including Norwegians of all ages. The purpose was to examine etiological factors for a wide

range of diseases. Overall participation rate of CONOR was 58% although it varied between the regional studies.

## Medical Birth Registry of Norway

The MBRN is a nationwide registry based on compulsory notification of all live- and stillbirths including home births from 16 weeks of gestation (12 week since 2002) [41]. It was established in 1967 and collects information such as maternal health before and during pregnancy, complications during pregnancy and delivery, interventions during labour, mother's and child's health during afterbirth and neonatal diagnoses through standardized forms. Information is registered by clinicians attending the mother and child and supplemented with data after birth until discharge. The birth notification form was changed in December 1998, check boxes were introduced for the most common conditions and some new information included, such as ultrasound-based estimation of gestational age.

## Eligibility criteria

We included fathers who had participated in at least one health surveys before the age of 50. If a father participated in more than one survey, we used the survey nearest to the birth of the child (referred here as 'the relevant health survey').

## Data linkage

Using the encrypted personal identification number–unique to each study participant –we linked information obtained from the health surveys to the Medical Birth Registry of Norway (MBRN) (Fig 1).

## Exposure

Self-reported information on paternal family history of premature stroke, defined in the relevant health survey as "history of stroke before 70 years in parents or siblings", and family history of premature CHD, defined in the relevant health survey as "history of myocardial infarct (MI) before 60 years in parents or siblings", were the exposure(s) in our study. Information on family history of stroke was not collected in all health surveys, thus the number of fathers with information on family history of stroke is lower than that of fathers with information on family history of CHD. (See Fig 1).

## Outcome

The outcomes were stillbirth (defined as late spontaneous abortions/ stillbirths from 16 weeks of gestation or more and where the foetus was born without signs of life), neonatal death (defined as death of the live born infant during the first 28 days of life) and perinatal death (defined as combination of stillbirth and neonatal death). The denominators for calculating stillbirth and perinatal death rates were total number of births from 16 weeks of gestation, while the total number of live births was used for calculating the neonatal death rate.

## Other variables

From health surveys, we obtained education, smoking, physical activity, height, and weight (used to calculate body mass index- BMI), non-fasting total cholesterol, systolic and diastolic blood pressure (used to define hypertension), non-fasting glucose levels, time since last meal and diabetic medication use (used to define diabetes).

**Family History of premature CHD**
(N=512,113)

- Screening of 40-year-old program (1985-1999)
  n = 380,206 (74.2%)
- Three County Health Survey (1974-1988)
  n = 54,686 (10.7%)
- Cohort of Norway (1994-2003)
  n = 77,219 (15.1%)
- Missing =12 (0.0%)

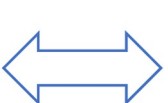

**Perinatal death**
Medical Birth Registry of
Norway
(1967-2020)
Total births N = 512,113
Stillbirth = 4,321 (0.8%)
Neonatal Death = 2,383 (0.5%)
Perinatal Death = 6,704 (1.3%)

**Family History of premature Stroke**
(N= 204,495) *

- Screening of 40-year-old program (1985-1999)
  n = 126,038 (61.6%)
- Three County Health Survey (1974-1988)
  n = 0
- Cohort of Norway (1994-2003)
  n = 77,219 (37.8%)
- Missing = 1,238 (0.6%)

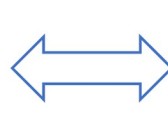

**Perinatal death**
Medical Birth Registry of Norway
(1967-2020)
Total births N= 204,495
Stillbirth = 1,371 (0.7%)
Neonatal Death = 682 (0.3%)
Perinatal Death = 2,053 (1.0%)

**Fig 1. Data sources, linkages and number of observations.** *Some health surveys do not collect family history of stroke at all.

Information obtained from the MBRN included paternal age [$\leq$ 25, 26–30 (reference), 31–39, $\geq$ 40 years] and maternal age at childbirth [$\leq$ 19, 20–24 (reference), 25–29, 30–34, $\geq$ 35 yr], number of births per father, marital status [married/cohabitation (reference), unmarried/single, widowed/divorced/ separated], birth year of child [1967–1977 (reference), 1978–1988, 1989–1999, 2000–2021], birth weight in grams, sex of child [male, female, unknown sex], gestational age [term: $\geq$ 37 completed weeks of gestation, preterm: < 37 completed weeks of gestation, extremely preterm: < 32 completed weeks of gestation], small for gestational age (SGA): birth weight by gestational age and sex less than 2.5 percentile (binary) [42], and congenital anomalies of child (binary).

When evaluating the association between family history of atherosclerotic disease and perinatal death, focusing on the total effect of paternal family history, paternal and maternal own risks factors such as hypertension, diabetes, total cholesterol, smoking, physical activity as well as infant factors such as birth weight, gestational age, SGA 2.5 percentile, foetal anomalies will be mediators (Fig 2), and should not be adjusted for. Nonetheless, to differentiate the effect of paternal family history from maternal known risk factors for perinatal death, we did additional

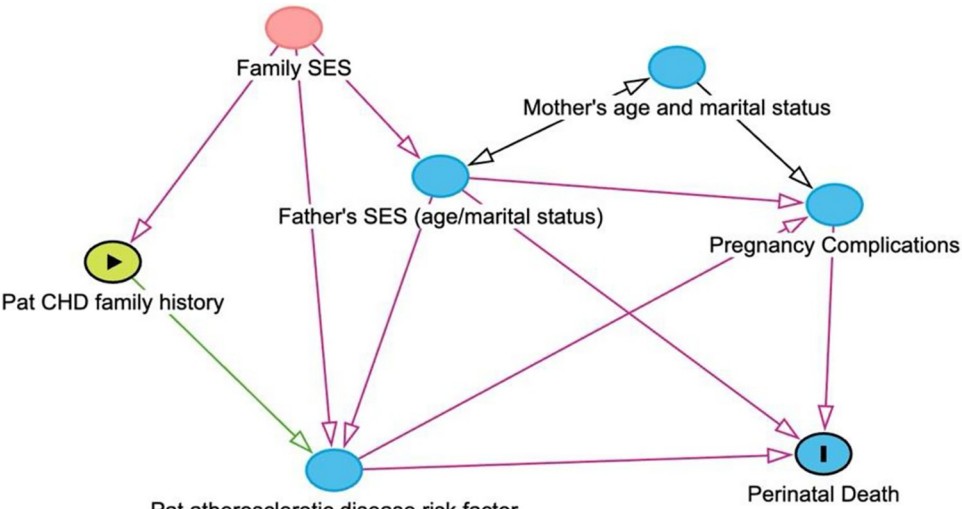

**Fig 2. Direct acyclic graphs for association between paternal history of premature atherosclerotic disease and perinatal death.** Pat CHD Family History: Paternal Family history of premature atherosclerotic disease (CHD and Stroke). Pat atherosclerotic disease risk factors: Paternal atherosclerotic risk factors (Diabetes, Hypertension, Total Cholesterol, Body Mass Index). Family SES: Family's socioeconomic status. Father's SES: Father's socioeconomic status. Pregnancy Complications: preeclampsia, small for gestational age less than 2.5th percentile, gestational age.

analyses where mothers with some important risk factors for perinatal death such as pre-eclampsia including HELLP syndrome and eclampsia, pregestational and gestational hypertension, and pregestational and gestational diabetes were excluded.

## Unit of analysis and study population

The birth was the observation unit in all analyses. We included singleton births where information on paternal family history of CHD or paternal family history of stroke was available from the health surveys in the analyses. We excluded multiple pregnancies/births since the causes of perinatal death may differ from that of singletons. Some fathers had more than one births and thus number of births did not represent number of fathers (S1 Table in S1 File).

## Statistical analyses

Continuous variables were presented with means and standard deviations (SD) and categorical variables as numbers and percentages (%). Differences in the father's characteristics across family history of premature atherosclerotic diseases (family history of CHD and Stroke), were tested using two sample t-test with equal variances for continuous variables and Pearson chi square test for categorical variables and corresponding p-values were reported. The analyses were conducted separately for CHD and Stroke since the number of fathers with information on each exposure differed.

For fathers with information on family history and more than one birth, their births could be registered with discrepant paternal family history because members of father's family experienced premature CHD or stroke between first birth and subsequent births. In addition, siblings conceived by the same father will be correlated. To address these issues, we chose to use multilevel mixed effects logistic regression with random intercepts by father's identification number to examine associations between paternal family history of atherosclerotic disease and perinatal death in offspring. Mixed models take into account and estimate the correlations between births to the same father. We reported odds ratios (OR) and corresponding 95%

confidence intervals (CI) with P-values < 0.05 considered statistically significant. Since still-birth, neonatal and perinatal death are all rare outcomes, odds will approximate risk in our analyses.

## Models

We built two models: *Model 1* adjusting for birth year of child, *Model 2* adjusting for birth year of child, paternal and maternal age at childbirth and paternal marital status. A likelihood-ratio tests comparing the models with ordinary logistic regressions were highly significant for all models. Variables in the final model (model 2) were chosen according to a Directed acyclic graph (DAG) developed using DAGitty (Fig 2) [43].

We adjusted for birth year of the child to address time trends as the changes in medical practices and lifestyle factors might influence the outcomes. According to our DAG, fathers' socioeconomic background is a potential confounder. However, we lacked this information. Adjustment for father's own education might have been relevant as a proxy for his socioeconomic background, but due to the high proportion of missing data (86.8% of fathers), we could not adjust for this either. Instead, we used marital status and age at childbirth which may capture some social factors, and we included these factors as proxy variables for father's socioeconomic background. We also conducted additional analyses for the subpopulation where information on father's education was available (13.2% of the study population). Statistical analyses were performed using Stata version 18.

## Missing data

There were no missing data on outcome and adjustment variables. There were 12 missing values for family history of CHD (0.0% on 512,113 of total observations) and 1,238 for family history of stroke (0.61% on 204,495 of total observations). The reason for having different number of observations between family history of CHD and stroke is that some surveys did not collect family history of stroke at all, and we thus had to use two separate study populations. Analyses were done separately for family history of CHD and family history of stroke and, we did complete case analyses in these two populations.

## Ethical considerations

The study was approved by the Regional Committee for Medical and Health Research Ethics, Norway (25.11.2019/34555). Health Survey participants have given written inform consent to use their data and to future connection to other data sources at the time of health survey and/or fall under the umbrella of national regulation law for national health surveys in Norway § 4–7 and § 7. The Regional Ethics committee exempted informed consent regarding registry data. Data were accessed on 16 August 2022.

## Results

In total, 220,386 fathers had information on family history of CHD and had 512,111 births. Of these fathers, 27,032 (12.3%) reported they had premature family history of CHD while 563 (0.3%) fathers reported discrepant family histories i.e., having different family history in first and subsequent births. Regarding stroke, 85,428 fathers had information on family history of premature stroke and had 203,257 births. Of these fathers, 7840 (9.2%) reported that they had premature family history of stroke while 13 (0.02%) fathers reported different family history in different pregnancies.

## Paternal characteristics

**Fathers with information on family history of CHD.**   Table 1 shows characteristics of fathers who had information on family history (the birth being the observation unit). Most births to fathers with information on family history of CHD occurred in the earlier time periods: 34.5% in 1967–77 and 42.9% in 1978–1988. Overall, the mean paternal age at childbirth was 29.9 years, mean BMI was 25.6 kg/m2 and mean non-fasting total cholesterol 5.8 mmol/dl. Overall, 38.3% of fathers were current smokers, 15.1% had a sedentary lifestyle and 34.4% had hypertension. The level of these risk factors was higher among father with family history of CHD compared to those without (Table 1).

## Fathers with information on family history of stroke

Most births to father with information on family history of stroke occurred in the earlier periods: 56.2% in 1967–1988, 33.8% in 1989–1999. The mean paternal age at childbirth was 30.5 years, most (73%) fathers were married or had a partner, their mean BMI was 26.1 kg/m2 and mean non-fasting total cholesterol was 5.9 mmol/l. Overall, 32.9% of fathers were current smokers, 9.4% had a sedentary lifestyle and 28.6% had hypertension. The level of these risk factors was again higher among fathers with family history of stroke compared to those without (Table 1). Of note, most fathers (89.4% for CHD and 85.4% for Stroke) participated in the health survey after their children were born. The mean (SD) time lag between health survey and childbirth was 10.0 (7.4) years and 9.0 (8.1) years for family history of CHD and stroke, respectively.

## Maternal characteristics

Maternal characteristics are displayed in S1.2 Table in S1 File. A total of 236,795 mothers delivered 512,111 births.

## Family history of CHD

The mothers whose partners had family history of CHD were slightly younger when delivering (26.7 vs 26.9 years) and had a slightly higher prevalence of preeclampsia (2.9% vs 2.6%) compared to those whose partners did not have family history of CHD, but differences were very small. Only a small proportion of them had chronic hypertension (0.2%), pregestational (0.2%) and gestational diabetes (0.2%), preeclampsia including HELLP syndrome and eclampsia (2.7%). Again, the differences between those with and without family history were small. Parity and the proportions of placenta previa and abruptio placenta were the same between the two group of mothers.

## Family history of stroke

Mothers whose partners had family history of stroke were slightly younger at the time of childbirth compared to their counterparts (27.3 vs 27.5 years). Parity, the proportions of mothers with chronic hypertension (0.3%), pre-gestational (0.3%) and gestational diabetes (0.4%), and pre-eclampsia (2.9), placental previa (0.2) and abruption (0.6) were almost the same in the groups with and without family history.

## Association between paternal family history of atherosclerotic disease and perinatal death

**Family history of premature CHD.**   Of births to fathers who reported a family history of CHD, 511 (0.8%), 271 (0.4%) and 782 (1.3%) were stillbirths, neonatal deaths, and perinatal

**Table 1. Characteristics of fathers: Overall and by fathers' family history of coronary heart disease or stroke ⊥.**

| Paternal Characteristics | Fathers with information on family history of CHD | | | Fathers with information on family history of stroke | | |
|---|---|---|---|---|---|---|
| | Total | No | Yes | Total | No | Yes |
| Number of fathers, n (%) | 220,386 (100) | 193,917 (87.9) | 27,032 (12.3) | 85,428 (100) | 77,601 (90.1) | 7,840 (9.2) |
| Number of births, n (%) | 512,111 (100) | 449,359 (87.8) | 62,752 (12.3) | 203,257 (100) | 184,550 (90.8) | 18,707 (9.2) |
| **Year of birth, n (%) | | | | | | |
| 1967–1977 (Reference) | 176,802 (34.5) | 159,704 (35.5) | 17,097 (27.3) | 15,642 (7.7) | 14,219 (7.7) | 1,423 (7.6) |
| 1978–1988 | 219,490 (42.9) | 189,466 (42.2) | 30,020 (47.8) | 98,473 (48.5) | 88,609 (48.0) | 9,864 (52.7) |
| 1989–1999 | 93,121 (18.2) | 79,793 (17.8) | 13,321 (21.2) | 68,662 (33.8) | 62,441 (33.8) | 6,221 (33.3) |
| 2000–2021 | 22,710 (4.4) | 20,396 (4.5) | 2,314 (3.7) | 20,480 (10.1) | 19,281 (10.5) | 1,199 (6.4) |
| *^No. births per father, n (%) | | | | | | |
| 1 | 36,763 (7.2) | 32,512 (7.2) | 4,251 (6.8) | 12,241 (6.02) | 11,161 (6.1) | 1080 (5.8) |
| 2–3 (Reference) | 374,707 (73.2) | 328,743 (73.2) | 45,964 (73.3) | 149,083 (73.4) | 135,289 (73.3) | 13,794 (73.7) |
| >4 | 100,641 (19.7) | 88,104 (19.6) | 12,537 (20.0) | 41,933 (20.6) | 38,102 (20.7) | 3,813 (20.5) |
| **Age at childbirth in year, mean (SD) | 29.9 (6.3) | 29.9 (6.3) | 29.7 (6.2) | 30.5 (6.3) | 30.5 (6.3) | 30.3 (6.4) |
| **Age at childbirth in year, n (%) | | | | | | |
| 18–25 | 137,651 (26.9) | 120,542 (26.8) | 17,109 (27.3) | 46,390 (22.8) | 41,834 (22.7) | 4,556 (24.4) |
| 26–30 (Reference) | 163,140 (31.9) | 143,025 (31.8) | 20,115 (32.1) | 62,653 (30.8) | 56,970 (30.9) | 5,683 (30.4) |
| 31–39 | 171,847 (33.7) | 150,527 (33.5) | 21,320 (34.0) | 76,825 (37.8) | 69,928 (37.9) | 6,897 (36.9) |
| 40–60 | 39,473 (7.71) | 35,265 (7.9) | 4,208 (6.7) | 17,389 (8.6) | 15,818 (8.6) | 1,571 (8.4) |
| **Marital Status at childbirth, n (%) | | | | | | |
| Married/Partner (Reference) | 413,829 (80.8) | 363,627 (80.9) | 50,194 (80.0) | 147,280 (72.5) | 133,317 (72.2) | 13,963 (74.6) |
| Unmarried | 47,626 (9.3) | 41,441 (9.2) | 6,181 (9.9) | 34,495 (17.00) | 31,948 (17.3) | 2,547 (13.6) |
| Widowed/Divorced/Separated | 47,304 (9.24) | 41,313 (9.2) | 5,991 (9.6) | 21,172 (10.4) | 19,000 (10.3) | 2,172 (11.6) |
| **Smoking, n (%) | | | | | | |
| Current | 196,087 (38.3) | 170,567 (38.0) | 25,520 (40.7) | 67,608 (33.3) | 60,780 (32.9) | 6,828 (36.5) |
| Ex-smoker | 135,563 (26.5) | 118,593 (26.4) | 16,970 (27.0) | 55,644 (27.4) | 50,528 (27.4) | 5,116 (27.4) |
| Never (Reference) | 175,505 (34.3) | 155,796 (34.7) | 19,709 (31.4) | 77,919 (38.3) | 71,358 (38.7) | 6,561 (35.1) |
| *^Physical Activity, n (%) | | | | | | |
| Sedentary | 77,236 (15.1) | 68,353 (15.2) | 8,883 (14.2) | 19,154 (9.4) | 17,352 (9.4) | 1,802 (9.6) |

*(Continued)*

**Table 1.** (Continued)

| Paternal Characteristics | | Fathers with information on family history of CHD | | | Fathers with information on family history of stroke | | |
|---|---|---|---|---|---|---|---|
| | | Total | No | Yes | Total | No | Yes |
| | Walking, cycling | 196,576 (38.4) | 175,129 (39.0) | 21,441 (34.2) | 39,567 (19.5) | 35,635 (19.3) | 3,932 (21.0) |
| | Light Sport (Reference) | 100,709 (19.7) | 89,857 (20.0) | 10,850 (17.3) | 16,775 (8.3) | 15,209 (8.2) | 1,566 (8.4) |
| | Hard Exercise | 12,231 (2.4) | 10,932 (2.4) | 1,299 (2.1) | 3,095 (1.5) | 2,796 (1.5) | 299 (1.6) |
| | Missing | 125,371 (24.5) | 105,088 (23.40) | 20,279 (32.3) | 124,666 (61.3) | 113,558 (61.5) | 11,108 (59.4) |
| **BMI, mean (SD) | | 25.6 (3.2) | 25.6 (3.1) | 25.9 (3.3) | 26.1 (3.3) | 26.1 (3.3) | 26.3 (3.4) |
| **BMI categories, n (%) | | | | | | | |
| | Normal (Reference) | 235,593 (46.0) | 208,636 (46.4) | 26,957 (43.0) | 81,321 (40.0) | 74,112 (40.2) | 7,209 (38.5) |
| | Overweight | 231,685 (45.2) | 202,500 (45.1) | 29,185 (46.5) | 99,815 (49.1) | 90,686 (49.1) | 9,129 (48.8) |
| | Obese | 43,147 (8.4) | 36,825 (8.2) | 6,322 (10.2) | 21,976 (10.8) | 19,622 (10.6) | 2,354 (12.6) |
| **Hypertension, n (%) | | | | | | | |
| | No | 335,254 (65.5) | 295,053 (65.7) | 40,201 (64.1) | 145,124 (71.4) | 132,233 (71.7) | 12,891 (68.9) |
| | Yes | 176,117 (34.4) | 153,680 (34.2) | 22,437 (35.8) | 58,069 (28.6) | 52,262 (28.3) | 5,807 (31.0) |
| **Total Cholesterol, mean (SD) | | 5.8 (1.1) | 5.1 (1.1) | 5.7 (1.1) | 5.9 (1.1) | 5.8(1.1) | 5.9 (1.1) |
| **Diabetes Mellitus, n (%) | | | | | | | |
| | No (Reference) | 212,981 (41.6) | 178,434 (39.7) | 34,547 (55.1) | 178,839 (88.0) | 162,128 (87.9) | 16,711 (89.3) |
| | Pre-diabetes | 6,164 (1.2) | 5,083 (1.1) | 1,081 (1.7) | 4,787 (2.4) | 4,284 (2.3) | 503 (2.7) |
| | Diabetes | 4,245 (0.8) | 3,670 (0.8) | 575 (0.9) | 2,141 (1.1) | 1,937 (1.1) | 204 (1.1) |
| | Missing | 288,721 (56.4) | 262,172 (58.3) | 26,549 (42.3) | 17,490 (8.6) | 16,201 (8.8) | 1,289 (6.9) |
| **Temporal relationship between participation in health survey and childbirth | | | | | | | |
| | Health Survey Before childbirth | 54,352 (10.6) | 47,518 (10.6) | 6,834 (10.9) | 29,743 (14.6) | 27,777 (15.1) | 1,966 (10.5) |
| | Health Survey After childbirth | 457,759 (89.4) | 401,841 (89.4) | 55,918 (89.1) | 173,514 (85.4) | 156,773 (85.0) | 16,741 (89.5) |
| **Time lag in year between health survey and childbirth, mean (SD) | | 10.0 (7.4) | 10.1 (7.4) | 9.6 (7.1) | 9.0 (8.1) | 8.9 (8.1) | 10.0 (7.3) |

Difference between those with or without family history were tested by two sample t-test or Pearson chi2 tests.

** P value<0.05 for both CHD and Stroke

*^ P value<0.05 for CHD and P value> 0.05 for Stroke

⊥Year and number of births, father's age and marital status at childbirth are from the Medical Birth Registry of Norway, remaining characteristics are from three population-based health surveys in Norway.

deaths, respectively (Table 2). The proportion of deaths was very similar among births to fathers without a family history. We did not find any association between a paternal family history of CHD and stillbirth (OR 1.01, 95% CI 0.92; 1.12), neonatal death (OR 0.98, 95%CI 0.86, 1.11) or perinatal death (OR 1.00, 95% CI 0.92, 1.08).

Table 2. Association between paternal family history of premature coronary heart disease and subsequent risk of perinatal death$^{\perp}$.

| Family history of premature CHD | | Total | Deaths | $^{\Sigma}$Odds Ratio (95% CI) | |
|---|---|---|---|---|---|
| | | | N (%) | Model 1* | Model 2** |
| Stillbirth | | | | | |
| | No | 449,359 | 3,810 (0.9) | 1.00 (Reference) | 1.00 (Reference) |
| | Yes | 62,752 | 511 (0.8) | 0.98 (0.89, 1.08) | 1.01 (0.92, 1.12) |
| Neonatal Death | | | | | |
| | No | 445,549 | 2,112 (0.5) | 1.00 (Reference) | 1.00 (Reference) |
| | Yes | 62,241 | 271 (0.4) | 0.97 (0.85, 1.11) | 0.98 (0.86, 1.11) |
| Perinatal Death | | | | | |
| | No | 449,359 | 5,922 (1.3) | 1.00 (Reference) | 1.00 (Reference) |
| | Yes | 62,752 | 782 (1.3) | 0.98 (0.90, 1.06) | 1.00 (0.92, 1.08) |

$^{\Sigma}$Multilevel mixed effect logistic regression with random intercepts by father's identification number. The birth is an observation unit.

*Adjusted for year of birth [1967–1977 (reference), 1978–1988, 1989–1999, 2000–2021]

**Adjusted for year of birth, paternal [$\leq$ 25, 26–30 (reference), 31–39, $\geq$ 40 yr] and maternal age [$\leq$ 19, 20–24 (reference), 25–29, 30–34, $\geq$ 35] at childbirth, marital status married/have partner (reference), unmarried, widowed/divorced/ separated]

$^{\perp}$Information on family history is from three population-based health surveys in Norway and information on perinatal deaths from the Medical Birth Registry of Norway.

**Family history of premature stroke.** Of births to fathers with a family history of stroke, 125 (0.7%) were stillbirths, 70 (0.4%) neonatal deaths and 195 (1.0%) perinatal deaths (Table 3). We did not find any association between a paternal family history of stroke and stillbirth (OR 1.00, 95% CI 0.82, 1.21), neonatal death (OR 1.09, 95%CI 0.84, 1.41) or perinatal death (OR 1.02, 95% CI 0.87, 1.20).

**Sensitivity analyses.** To disentangle the effect of paternal family history from maternal known risk factors for perinatal death, we repeated analyses after excluding mothers with some important risk factors for perinatal death (preeclampsia including HELLP syndrome and eclampsia, pregestational and gestational hypertension, and pregestational and gestational

Table 3. Association between paternal family history of premature stroke and subsequent risk of perinatal death$^{\perp}$.

| Family history of premature stroke | | Total | Deaths | $^{\Sigma}$Odds Ratio (95% CI) | |
|---|---|---|---|---|---|
| | | | N (%) | Model 1* | Model 2** |
| Stillbirth | | | | | |
| | No | 184,550 | 1,246 (0.7) | 1.00 (Reference) | 1.00 (Reference) |
| | Yes | 18,707 | 125 (0.7) | 1.00 (0.82, 1.23) | 1.00 (0.82, 1.21) |
| Neonatal Death | | | | | |
| | No | 183,304 | 612 (0.3) | 1.00 (Reference) | 1.00 (Reference) |
| | Yes | 18,582 | 70 (0.4) | 1.09 (0.85, 1.41) | 1.09 (0.84, 1.41) |
| Perinatal Death | | | | | |
| | No | 184,550 | 1,858 (1) | 1.00 (Reference) | 1.00 (Reference) |
| | Yes | 18,707 | 195 (1) | 1.02 (0.87, 1.20) | 1.02 (0.87, 1.20) |

$^{\Sigma}$Multilevel mixed effect logistic regression with random intercepts by father's identification number. The birth is an observation unit.

*Adjusted for year of birth [1967–1977 (reference), 1978–1988, 1989–1999, 2000–2021]

**Adjusted for year of birth, paternal [$\leq$ 25, 26–30 (reference), 31–39, $\geq$ 40] and maternal age [$\leq$ 19, 20–24 (reference), 25–29, 30–34, $\geq$ 35] at childbirth in years, marital status married/have partner (reference), unmarried, widowed/divorced/ separated]

$^{\perp}$Information on family history is from three population-based health surveys in Norway and information on deaths from the Medical Birth Registry of Norway, 1967–2021.

**Table 4. Characteristics of perinatal deaths by father's family history of premature atherosclerotic disease [⊥].**

| Perinatal deaths Characteristics | Fathers with information on family history of CHD | | | Fathers with information on family history of stroke | | |
|---|---|---|---|---|---|---|
| | Total | No | Yes | Total | No f_ | Yes |
| Number of dead infants, n (%) | 6,704 (100) | 5,922 (88.3) | 782 (11.7) | 2,053 (100) | 1,858 (90.5) | 195 (9.5) |
| Birthweight (g), mean (SD) | 1782 (1233) | 1791 (1232) | 1714 (1236) | 1,662 (1277) | 1,655 (1277) | 1725 (1273) |
| Sex, n (%) | | | | | | |
| Male (Reference) | 3,694 (55.1) | 3,262 (55.1) | 432 (55.2) | 1,148 (55.9) | 1,042 (56.1) | 106 (54.4) |
| Female | 2,838 (42.3) | 2,510 (42.4) | 328 (41.9) | 820 (39.9) | 738 (39.7) | 82 (42.1) |
| Unknown sex | 172 (2.6) | 150 (2.5) | 22 (2.8) | 85 (4.1) | 78 (4.2) | 7 (3.6) |
| Gestational Age, n (%) | | | | | | |
| Term (Reference) | 2,168 (36.8) | 1,930 (37.0) | 238 (35.2) | 603 (35.0) | 538 (34.7) | 65 (37.8) |
| Preterm | 2,103 (35.7) | 1,857 (35.6) | 246 (36.4) | 534 (31.0) | 484 (31.2) | 50 (29.1) |
| Extremely Preterm | 1,615 (27.4) | 1,423 (27.3) | 192 (28.4) | 586 (34.0) | 529 (34.1) | 57 (33.1) |
| SGA 2.5 percentile, n (%) | | | | | | |
| No (Reference) | 4,451 (75.9) | 3,951 (76.0) | 500 (74.5) | 1,353 (78.7) | 1,223 (78.9) | 130 (76.5) |
| Yes | 1,416 (24.1) | 1,245 (24.0) | 171 (25.5) | 367 (21.3) | 327 (21.1) | 40 (23.5) |
| Congenital Anomalies, n (%) | | | | | | |
| No (Reference) | 5,529 (82.5) | 4,895 (82.7) | 4,895 (81.1) | 1,677 (81.7) | 1,511 (81.3) | 166 (85.1) |
| Yes | 1,175 (17.5) | 1,027 (17.3) | 148 (18.9) | 376 (18.3) | 347 (18.7) | 29 (14.9) |

Difference between those with or without family history were tested by two sample t-test or Pearson chi2 tests. None of them was different significantly.

[⊥]All these information was obtained from the Medical Birth Registry of Norway

diabetes). The resulting point estimates and 95% CI were almost the same as in the total population (S1.3 and S1.4 Tables in S1 File).

We also conducted separate analyses for the subpopulation where information on father's education was available (13.2% of the study population). The results of the analyses when adjusting for paternal education, birth year of child, paternal and maternal age at childbirth in this subgroup, gave almost the same estimates and 95% CI as when the analyses were adjusted for marital status, birth year of child, paternal and maternal age at childbirth (S1.5 and S1.6 Tables in S1 File).

**Characteristics of the perinatal losses by father's family history of premature atherosclerotic disease.** Table 4 describes characteristics of the perinatal deaths. Perinatal deaths in fathers with family history of CHD had lower mean birth weight than those to fathers without family history (1710 g vs 1791 g), and more of the births were preterm and extremely preterm. Likewise, SGA (25.5% vs 24.0%) and congenital anomalies (18.9% vs 17.3%) were more frequent in fathers with family history of CHD. However, none of these differences were statistically significant.

In contrast, perinatal deaths to fathers with a family history of stroke had higher mean birth weight than those to fathers without a family history of stroke (1725 g vs 1655g). There were also less preterm (29% vs 31.2%) and extremely preterm delivery (33.1% vs 34.1%) among the perinatal deaths to fathers with than without a family history. Congenital anomalies were also registered less frequently among those whose fathers had a family history (14.9% vs 18.7%). On the other hand, SGA (25th percentile) was more frequent (23.5% vs 21.1%). Numbers of

deaths to fathers with family history of premature stroke were low, however, and again none of these differences were statistically significant.

## Discussion

Using a large cohort of half a million births, we explored the potential association of paternal family history of atherosclerotic diseases with offspring perinatal death. We did not find an association between neither family history of premature CHD nor stroke and stillbirth, neonatal death, or perinatal death, despite less favourable risk factor profiles among fathers with a family history.

To the best of our knowledge, no previous study has addressed the association between paternal family history of atherosclerotic disease and perinatal death. However, our finding of no association is in line with the results of another population based Norwegian cohort study that examined the role of paternal genetic liability for CHD and adverse pregnancy outcomes such as miscarriage, stillbirth and SGA [44]. This study reported no association between paternal genetic risk scores that measures the genetic susceptibility for CHD and adverse pregnancy outcomes while maternal genetic CHD risk was associated with increased risk of hypertensive disorders of pregnancy, and SGA.

A few studies that examined maternal family history and pregnancy loss reported inconsistent results. A prospective cohort study of 742 pregnant women from United States found no association between self-reported maternal family history of cardiovascular (CVD) or premature CVD in first-degree relatives and risk of pregnancy loss [37]. A more recent population-based cohort study conducted in Norway, analysed 17,320 births and reported no association between maternal family history of MI and stillbirth (IRR 0.57, 95% CI 0.18, 1.83) [36]. However, the same study found an association between family history of stroke and increased risk of stillbirth (IRR 2.53, 95% CI 1.06, 6.01). Although not directly comparable to our study, two other cohort studies from Scotland [31] and Denmark [32] reported higher risk of mortality or hospitalization due to CHD or stroke in the parents of women with repeated history of miscarriages highlighting possible common mechanisms between pregnancy loss and atherosclerotic diseases in both heart and brain.

Taken together, the published literature suggests that although family history reflects both genetic susceptibilities, shared environment, and common behaviours, it may be possible that genetic factors following the fathers have different implications for perinatal death than genetic factors following the mothers. Genetic factors acting through mothers will include both mitochondrial genes and maternal genes which act on placental factors, the intrauterine environment and a variety of pregnancy complications in addition to contributing to the foetal genes [45]. Genetic factors acting through fathers, on the other hand, would be mainly through the contribution to foetal genes transmitted from both mother and father [45]. Epidemiological studies have also shown that the risk of pregnancy complications through affected mothers are higher than that of the risk through fathers [22, 24, 44, 45].

There also are other possible explanations for the current findings. Because of the multi-faceted nature of pregnancy loss and perinatal death, it is also possible that predisposition to atherosclerotic disease may only be related to specific subset of losses for example recurrent pregnancy loss or early pregnancy loss (earlier than 12 week of gestation). A non-differential misclassification of family history could also reduce the effect size (see the limitations section). Our data have excellent power to detect effect size as large as 1.25 or greater. However, for small effect sizes, we might also have lacked power.

Father's age at childbirth is an acknowledged risk factor for stillbirth and miscarriage with risk increase related to high age [10]. According to our DAG, it is also a mediator between

family history and perinatal loss. In our study there was minimal difference in fathers' mean age at childbirth between those with and without family history [29.7 years (SD 6.2) vs 29.9 years (SD 6.3) for family history of CHD and 30.3 years (6.4) vs 30.5 years (6.3) for family history of stroke]. Also, the proportions of fathers who were 40+_years, the age that associates with sperm DNA strand breaks, genetic imprinting errors and chromosomal anomalies [10, 46] were even lower in those with family history than those without (6.7% vs 7.9% for family history of CHD and 8.4% vs 8.6% in family history of stroke).

Similar findings were found for BMI, another acknowledge risk factor for aberrant foetal growth and pregnancy complications [47–50]. Mean (SD) of BMI among fathers with family history and without family history were 25.9 kg/m$^2$ (3.3) vs 25.6 (3.1) in CHD and 26.3 (3.4) vs 26.1 (3.3) in stroke. In case of other probable mediating risk factors, although there were differences in proportions among those with and without family history, the differences were small. For example, 35.8% vs 34.2% in case of hypertension and 40.7% vs 38.0% in smoking. The fathers might have been even healthier at the time of childbirth since almost 90% of these risk factors were measured after their childbirth with an average time lag of 10 years.

Regarding maternal characteristics the differences between those whose partners had or did not have family history (for instances age, hypertension and diabetes) were small However, maternal age was adjusted in model 2 and we did additional analyses excluding mothers with some important risk factors for perinatal loss, and results were basically the same.

Despite the finding of no associations between paternal family history and offspring perinatal death, we evaluated characteristics among the perinatal deaths to fathers with and without family histories. While the perinatal deaths to fathers with a family history of CHD had lower mean birth weight and were more likely to be preterm, extremely preterm, and SGA than those whose fathers did not have a family history of CHD, the opposite pattern was found when comparing perinatal deaths to fathers with and without a family history of stroke. However, none of the differences were statistically significant, and these findings need to be replicated in larger studies.

## Strengths of the study

We examined paternal history of premature atherosclerotic disease and its association with perinatal death using information from a large population of males in their reproductive age in Norway, linked to a high-quality data source with national data on births and perinatal outcomes through a personal identification number, unique to each study participant.

## Limitations

When interpreting our findings, one needs to keep in mind certain limitations, inherent to both population and register-based studies. Since fathers chose whether to participate in the health surveys, a potential self-selection bias must be considered. Having a family history of CVD might be a motivation for fathers to participate in a health survey where one focus was CVD. However, since CVD in a father's family was (and is) not an acknowledged risk factor for perinatal loss, participation is likely not related to both a family history of CVD and the experience of a perinatal loss. The prevalence of perinatal loss in the total study population was also similar to that in the total Norwegian population during the study years [51]. Moreover, even in a situation with self-selection into the study population, this does not necessarily affect the relation between a given exposure and a given outcome. A study evaluating selection bias in the Mother, Father and Child cohort study in Norway, concluded that in spite of a participation rate around 40%, higher socioeconomic level in participants than non-participants with

bias in prevalence estimates of exposures and outcomes, the studied exposure-outcome associations were valid [52].

Further, misclassification of exposure is another potential limitation. The relationship between paternal family history of atherosclerotic diseases and perinatal death was unknown at the time of survey and thus this misclassification is likely non-differential [53] which could attenuate our results. A large non-differential misclassification could then be a factor explaining our null findings. Errors in reporting of family history can occur when individuals are unaware of their relatives' disease status or when their relatives themselves are unaware of their own health status, thus leading to misclassification of the exposure [54]. However, a study that investigates concordance of family history of CVD between offspring and parents found that family history provided by young and middle-aged Norwegian stroke patients is in good concordance with parental report of their own CVD status [55]. A high proportion of missing in family history could also lead to misclassification. However, we have a small amount of missing data (less than 1%) in both family history of CHD and stroke. To sum up, we can conclude that the potential non-differential misclassification of family history in our study is likely small and not enough to explain our null findings.

Regarding imprecision, the confidence intervals are relatively narrow when looking at family history of CHD due to the large study population, and imprecision should not be a problem. The study population is smaller when looking at family history of stroke, with somewhat wider confidence intervals and somewhat lower precision. However, point estimates were close to 1.0 and results agree with those found for family history of CHD, and we believe that the point estimates are likely valid.

For studying the effect of paternal family history on perinatal death, it is relevant to adjust for the fathers' socioeconomic background. Unfortunately, we lacked that information and had large proportions of missing data for father's own education, a proxy for his socioeconomic background. We did, however, have information on marital status (being married or having a partner) at the time of childbirth and age at childbirth which may capture some social factors, and we therefore included these as proxy variables for father's socioeconomic background. We also conducted separate analyses for the subpopulation where information on father's education was available and got almost the same estimates and 95% CI when adjusting for father's age and education as when adjusting for father's age and marital status. Therefore, we argue that using age and marital status is a satisfactory way to deal with missing information on father's education. Nonetheless, we still cannot exclude residual confounding by other unknown factors.

## Implications of findings and future result directions

Family history gives information about genetic and persisting environmental risk factors for many diseases. Taking a family history has low cost and is an acceptable and informative first step to identify individuals from the general population who may benefit from genetic testing. Some studies found association between maternal family history of CVD and pregnancy loss or stillbirth. Although we did not find association between paternal history of premature atherosclerotic disease and perinatal death, we did find that fathers with a family history of premature atherosclerotic disease were more likely to smoke, be obese and have higher blood pressure and higher cholesterol than fathers without a family history. A study reported that the risk of pregnancy loss increased with increasing number of cardiovascular risk factors in fathers. Our study was not able to relate these factors to the risk of perinatal loss since most of the fathers measured these factors in health surveys after the birth, often with a long-time lag. It does, however, suggest that family history of premature atherosclerotic disease might be an

indicator of the fathers' underlying health that could be optimised during preconception period. It also calls for more studies where the relation between these cardiovascular risk factors in fathers can be studied in relation to future risk of perinatal loss.

We also found that infants born to fathers with family history of CHD had lower mean birth weight, were more likely to be born preterm, extremely preterm, being SGA and more likely to have congenital anomalies than those without family history. We therefore warrant larger population-based studies to replicate and confirm the findings of the current study, also in low- and middle-income countries where both atherosclerotic risk factors and perinatal deaths are higher than in the Norwegian population. Further studies on how the experience of a perinatal death may affect fathers' health and risk factors later in life would be of interest as well. The perinatal care clinicians could collect information on family history of premature atherosclerotic disease in both parents to identify those who might benefit from closer follow up during pregnancy.

## Conclusions

We found that paternal family history of premature CHD or stroke did not associate with perinatal death in our large nationwide sample of 220, 390 fathers and 512 111 births. We did, however, find that fathers with a family history or premature atherosclerotic disease were less healthy than fathers without a family history and thus taking a family history could be a cost-effective screening tool for preconception and perinatal care physicians to access fathers' underlying health that may affect pregnancy outcomes. Larger population-based studies where there is little risk of self-selection and misclassification bias that can investigate the relation between CVD risk factors and perinatal loss in future pregnancies would be of interest.

## Supporting information

**S1 File. Supplementary tables.**
(DOCX)

**S2 File. Regression outputs.**
(DOCX)

## Acknowledgments

We thank Tatiana Fomina for Data Curation, Magne Haugland Solheim, Jannicke Igland, and Rolv Terje Lie for advice on statistical analyses.

## Author Contributions

**Conceptualization:** Kari Klungsøyr, Grace M. Egeland.

**Data curation:** Gerhard Sulo.

**Formal analysis:** Shwe Sin Win.

**Funding acquisition:** Kari Klungsøyr, Grace M. Egeland.

**Methodology:** Kari Klungsøyr, Gerhard Sulo.

**Supervision:** Kari Klungsøyr, Gerhard Sulo.

**Writing – original draft:** Shwe Sin Win.

**Writing – review & editing:** Shwe Sin Win, Kari Klungsøyr, Grace M. Egeland, Gerhard Sulo.

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
