## [Decision Letter · Decision Letter 0]

23 Jul 2024

PONE-D-24-10471Paternal family history of premature atherosclerotic disease and perinatal death: a population-based cohort study

PLOS ONE

Dear Dr. Win,

Thank you for submitting your manuscript to PLOS ONE. After careful consideration, we feel that it has merit but does not fully meet PLOS ONE’s publication criteria as it currently stands. Therefore, we invite you to submit a revised version of the manuscript that addresses the points raised during the review process.

This is a largely well written and interesting manuscript. Both reviewers have suggested changes to the manuscript that I believe will improve it. Whilst all the reviewers' suggestions need dealing with, reviewer 1's point about the Discussion section where they say "Provide a more detailed analysis of the potential reasons for the lack of significant associations" (point 1 about the Discussion section) and reviewer 2's point "Even after reading your study, I'm doubtful in terms of pathophysiology of the link that you tried to highlight between paternal family history of atherosclerotic diseases and offspring perinatal death" need particular attention in order for the manuscript to be considered suitable for publication.

In addition to the points raised by the reviewers, the authors need to make the manuscript more compliant with STROBE guidelines for observational studies (Submission Guidelines | PLOS ONE; Checklists - STROBE (strobe-statement.org); http://record-statement.org). Changes that need to be made in this context include:

1. An acknowledgment of the limitations of the study in the abstract.

2. Eligibility criteria for the different cohort participants that were used for this study need to be added.

3. Describe efforts to address potential sources of bias.

4. How were missing data addressed?

5. Discuss limitations of the study in more detail, taking into account sources of potential bias or imprecision. Discuss both direction and magnitude of any potential bias.

We look forward to receiving your revised manuscript.

Kind regards,

Clive J. Petry, PhD

Academic Editor

PLOS ONE

Reviewers' comments:

Reviewer's Responses to Questions

**Comments to the Author**

1. Is the manuscript technically sound, and do the data support the conclusions?

Reviewer #1: Yes

Reviewer #2: Yes

2. Has the statistical analysis been performed appropriately and rigorously? 

Reviewer #1: Yes

Reviewer #2: Yes

3. Have the authors made all data underlying the findings in their manuscript fully available?

Reviewer #1: No

Reviewer #2: Yes

4. Is the manuscript presented in an intelligible fashion and written in standard English?

Reviewer #1: Yes

Reviewer #2: Yes

5. Review Comments to the Author

Reviewer #1: Review on the manuscript “Paternal family history of premature atherosclerotic disease and perinatal death: a population-based cohort study”

Thank you very much for providing me the opportunity to review this manuscript. The study provides a detailed overview of a population-based cohort study investigating the association between paternal family history of premature atherosclerotic disease and perinatal death. The study fills a gap in the literature regarding non-modifiable paternal risk factors and perinatal death, contributing valuable insights to the field.

Abstract: Well-written

Introduction:

The introduction effectively convey the information by providing a comprehensive background on perinatal mortality, defining key terms, and discussing the multifactorial causes of perinatal death. It also highlights the significance of paternal factors, particularly genetic influences, which have been less studied compared to maternal factors. The identification of a research gap concerning the association between paternal family history of premature atherosclerotic diseases and perinatal outcomes is clear and compelling.

However, the introduction could be improved by breaking down the information into more concise sections or paragraphs to enhance readability. There is a need to focus more on the study's rationale and objectives to make it more engaging. Incorporating a brief summary of the key points at the end could help reinforce the study's purpose and significance.

Materials and Methods Review

The "Materials and Methods" section of the study is comprehensive and well-structured, providing clear descriptions of data sources, exposures, outcomes, and statistical analyses. Nevertheless, there are some weaknesses which can improve the readability and quality of the manuscript.

1. At some places provided information seems to be dense and might be overwhelming for some readers. Breaking it into more concise paragraphs and adding subheadings could improve readability. For example, introduce subheadings for sections like "Data Sources," "Linkage and Inclusion Criteria," "Definitions," "Statistical Analysis," and "Ethical Considerations." This will make the section more navigable.

2. The data spans from 1974 to 2003; changes in medical practices and lifestyle factors over these decades might influence the outcomes and their interpretation in the current context.

3. The lack of socioeconomic data and high missing rates for father's education are noted as limitations. Addressing how these missing data might impact the study's findings could provide a more balanced view.

4. Incorporate visual aids, such as flowcharts or diagrams (like the directed acyclic graph mentioned), to illustrate data linkages and the analysis process. This can help in understanding complex methodologies.

Results

The "Results" section presents a clear and comprehensive overview of the findings from the study. However, there are some areas for improvement to enhance clarity and readability.

1. The section lacks detailed explanations and interpretations of the observed differences in maternal characteristics. Provide more interpretation and discussion on the observed differences in maternal characteristics and their potential implications for the study's findings.

2. The section provides limited context on why the maternal characteristics are important in relation to the study's main focus on paternal family history and perinatal death.

3. The section lacks interpretation of the findings, such as why there may be no significant association between paternal family history of CHD and perinatal death.

4. The reporting of percentages and means could be clearer and more consistent to improve readability. Ensure consistent and clear reporting of percentages and means, and consider using additional visual aids to improve clarity.

Discussion

The discussion section effectively summarizes the key findings of the study, situating them within the broader context of existing research. However, there are several areas where the discussion could be strengthened to enhance clarity, depth, and engagement.

1. The discussion lacks in-depth analysis and interpretation of why the expected associations were not observed and what this implies for future research. Provide a more detailed analysis of the potential reasons for the lack of significant associations. Discuss the possible impact of measurement timing, residual confounding, and misclassification of family history.

2. The discussion could benefit from a deeper engagement with the broader implications of the findings, especially concerning genetic and environmental factors. Elaborate on the broader implications of the findings for understanding genetic and environmental contributions to perinatal outcomes. Discuss how these results can inform future research directions.

3. The flow could be improved by better organizing the comparison with other studies and making clearer transitions between different sections of the discussion. Reorganize the discussion to improve coherence. Start with a summary of the key findings, followed by a comparison with existing literature, then move on to strengths and limitations, and end with broader implications and future directions.

Conclusion

The conclusions section is well-written. However, there are areas where the conclusion can be enhanced to provide a more comprehensive and impactful summary.

1. The conclusion could benefit from more depth and specificity regarding the implications of the findings and the observed risk factors. Provide more detail on the implications of the findings for clinical practice and public health, especially regarding paternal health screening and intervention.

2. Place the findings in a broader context by discussing their relevance to other studies and the current understanding of perinatal health risks.

3. If applicable, suggest any policy recommendations or changes in clinical practice that could arise from the findings.

Reviewer #2: 3 comments

-Until 1975, for the WHO, the limit of viability was 1000 g and 28 weeks. After this date, the theoretical definition of a birth living was the following: “the complete expulsion or extraction of the body of the mother of a product of conception weighing at least 500 grams (or a gestational age of 22 years weeks). Lines 99-101?

-Model 2 adjusting for birth year of child, paternal and maternal age at childbirth and paternal marital status. Why paternal marital status? Marital status is an unsuitable proxy for socio-economic factor

-Even after reading your study, I'm doubtful in terms of pathophysiology of the link that you tried to highlight between paternal family history of atherosclerotic diseases and offspring perinatal death

But the study is easy to read and well structured

6. PLOS authors have the option to publish the peer review history of their article (what does this mean?). If published, this will include your full peer review and any attached files.

Reviewer #1: **Yes: **Rubeena Zakar

Reviewer #2: **Yes: **Félicia Joinau-Zoulovits

---

## [Author Response · Author response to Decision Letter 0]

24 Sep 2024

We reply to each comment in point-by-point fashion. Our responses are shown as “Response”. Page and line numbers in the rebuttal letter correspond to the document named “Revised Manuscript with Track Changes” while all tracked changes are shown.

1. An acknowledgment of the limitations of the study in the abstract.

Response: Thank you for pointing this out. Major limitations of the study were added to the abstract. See Page 3, Line 51-52: 

“Limitations: Socioeconomic background of fathers was captured using imperfect proxy. Potential misclassification of family history and selection bias should be considered.” 

2. Eligibility criteria for the different cohort participants that were used for this study need to be added.

Response: See Page 12, Line 269-272 

“Eligibility criteria are fathers who had participated in at least one health survey before the age of 50. If a father participated in more than one survey, we used the survey nearest to the birth of the child (referred to here as ‘the relevant health survey’)”

3. Describe efforts to address potential sources of bias.

Response: Possible sources of bias in our study as mentioned in limitations (Page 39-42, Line 694-759) are selection bias infringed in the traditional survey research, information bias due to misclassification of having family history or not and residual confounding. 

See page 39-40, lines 692-703 where we have discussed potential selection bias:

“Since fathers chose whether to participate in the health surveys, a potential self-selection bias must be considered. Having a family history of CVD might be a motivation for fathers to participate in a health survey where one focus was CVD. However, since CVD in a father’s family was (and is) not an acknowledged risk factor for perinatal loss, participation is likely not related to both a family history of CVD and the experience of a perinatal loss. The prevalence of perinatal loss in the total study population was also similar to that in the total Norwegian population during the study years.52 Moreover, even in a situation with self-selection into the study population, this does not necessarily affect the relation between a given exposure and a given outcome. A study evaluating selection bias in the Mother, Father and Child cohort study in Norway, have concluded that in spite of a participation rate around 40%, higher socioeconomic level in participants than non-participants with bias in prevalence estimates of exposures and outcomes, the studied exposure-outcome associations were valid.53”

See page 41, lines 717-724 where we discussed potential misclassification bias: 

“Further, misclassification of exposure is another potential limitation. The relationship between paternal family history of atherosclerotic diseases and perinatal death was unknown at the time of survey and thus this misclassification is likely non-differential54 which could attenuate our results. A large non-differential misclassification could then be a factor explaining our null findings. Errors in reporting of family history can occur when individuals are unaware of their relatives' disease status or when their relatives themselves are unaware of their own health status, thus leading to misclassification of the exposure.55 However, a study that investigates concordance of family history of CVD between offspring and parents found that family history provided by young and middle-aged Norwegian stroke patients is in good concordance with parental report of their own CVD status.56 A high proportion of missing in family history could also lead to misclassification. However, we have a small amount of missing data (less than 1%) in both family history of CHD and stroke. To sum up, we can conclude that the potential non-differential misclassification of family history in our study is likely small and not enough to explain our null findings.”

See page 42, lines 742-753 where we discussed residual confounding due to missing data on socioeconomic level:

“For studying the effect of paternal family history on perinatal death, it is relevant to adjust for the fathers' socioeconomic background. Unfortunately, we lacked that information and had large proportions of missing data for father’s own education, a proxy for his socioeconomic background. We did, however, have information on marital status (being married or having a partner) at the time of childbirth and age at childbirth which may capture some social factors, and we therefore included these as proxy variables for father’s socioeconomic background. We also conducted separate analyses for the subpopulation where information on father’s education was available and got almost the same estimates and 95% CI when adjusting for father’s age and education as when adjusting for father’s age and marital status. Therefore, we argue that using age and marital status is a satisfactory way to deal with missing information on father’s education. Nonetheless, we still cannot exclude residual confounding by other unknown factors.”

4. How were missing data addressed?

Response: To address editor’s concern on missing data, we added a section on missing data in the methods. Page 18, Line 395-403: 

“There were no missing data on outcome and adjustment variables. There were 12 missing values for family history of CHD (0.0 % on 512,113 of total observations) and 1,238 for family history of stroke (0.61% on 204,495 of total observations). The reason for having different number of observations between family history of CHD and stroke is that some surveys did not collect family history of stroke at all, and we thus had to use two separate study populations. Analyses were done separately for family history of CHD and family history of stroke, and we did complete case analyses in these two populations.”

5. Discuss limitations of the study in more detail, taking into account sources of potential bias or imprecision. Discuss both direction and magnitude of any potential bias.

Response: In the revised version, we expanded the ‘Limitations’ (Page 39-42, Line 687-753) to accommodate the Editor’s concerns with regards to sources of bias and imprecision. See page 39-40, lines 692-703 where we have discussed potential selection bias, see page 41, lines 717-734 where we discussed potential misclassification bias, see page 42, lines 742-753 where we discussed residual confounding due to missing data on socioeconomic level, see page 41, line 735-740 where we discussed imprecision. 

Reviewer #1: Review on the manuscript “Paternal family history of premature atherosclerotic disease and perinatal death: a population-based cohort study”

Thank you very much for providing me the opportunity to review this manuscript. The study provides a detailed overview of a population-based cohort study investigating the association between paternal family history of premature atherosclerotic disease and perinatal death. The study fills a gap in the literature regarding non-modifiable paternal risk factors and perinatal death, contributing valuable insights to the field. 

Abstract: Well-written

Introduction:

The introduction effectively conveys the information by providing a comprehensive background on perinatal mortality, defining key terms, and discussing the multifactorial causes of perinatal death. It also highlights the significance of paternal factors, particularly genetic influences, which have been less studied compared to maternal factors. The identification of a research gap concerning the association between paternal family history of premature atherosclerotic diseases and perinatal outcomes is clear and compelling. However, the introduction could be improved by breaking down the information into more concise sections or paragraphs to enhance readability. There is a need to focus more on the study's rationale and objectives to make it more engaging. Incorporating a brief summary of the key points at the end could help reinforce the study's purpose and significance.

Response: Based on the reviewer comment, we broke down the introduction into concise paragraphs and highlighted the study’s rationale and objectives. A summary of key point was also added at the end (Page 9 Line 193-205). 

“To sum up, previous studies have documented associations between several sociodemographic, lifestyle and environmental factors in fathers and the risk of perinatal loss. Further, paternal factors have been found to influence the risk of pregnancy complications associated with perinatal loss. The risk of pregnancy loss increased with increasing number of cardiovascular risk factors in fathers, this could be explained by common underlying aetiologic factors. Family history of cardiovascular disease reflects genetic susceptibilities, shared environment, and common behaviors and may be used to highlight potential common underlying aetiologic mechanisms for both cardiovascular risk and perinatal loss. So far, no studies have focused paternal family history of cardiovascular disease and the risk of perinatal loss. This study, therefore, aimed at exploring the potential association between paternal family history of premature coronary heart disease or stroke and stillbirth, neonatal death, and perinatal death that may share common underlying aetiologic mechanisms.” 

Materials and Methods Review

The "Materials and Methods" section of the study is comprehensive and well-structured, providing clear descriptions of data sources, exposures, outcomes, and statistical analyses. Nevertheless, there are some weaknesses which can improve

the readability and quality of the manuscript.

1. At some places provided information seems to be dense and might be overwhelming for some readers. Breaking it into more concise paragraphs and adding subheadings could improve readability. For example, introduce subheadings for sections like "Data Sources," "Linkage and Inclusion Criteria," "Definitions," "Statistical Analysis," and "Ethical Considerations." This will make the section more navigable.

Response: Thank you for pointing this out. Subheadings are added as suggested in methods. 

2. The data spans from 1974 to 2003; changes in medical practices and lifestyle factors over these decades might influence the outcomes and their interpretation in the current context.

Response: We totally agree with reviewer that medical practices and lifestyle factors have changed over these decades, possibly leading to better care and reduction of perinatal mortality. We have therefore adjusted all analyses for year of birth (also the “crude” analyses – Model 1) to avoid the potential temporal effect of these changes in our associations (Page 17, line 374-375). Further, we tested whether there is an interaction between exposure and year of childbirth and did not find one (P for interaction = 0.531)

3. The lack of socioeconomic data and high missing rates for father's education are noted as limitations. Addressing how these missing data might impact the study's findings could provide a more balanced view. 

Response: To address the reviewer’s concern, we conducted separate analyses for the subpopulation where information on father’s education was available (13.2% of the study population) and updated in the method section. Page 17, Line 390-393.

The results of the analyses (adjusting for age at birth and paternal education) in this subgroup, gave almost the same estimates and 95% CI as when the analyses were adjusted for age at birth and marital status (as done in the main analyses). Page 29, Line 542-547 and tables at Supporting Information1, Table S5 and S6. 

We have also added a comment on this in the Discussion part, Limitations, page 42, lines 742-753. 

“For studying the effect of paternal family history on perinatal death, it is relevant to adjust for the fathers' socioeconomic background. Unfortunately, we lacked that information and had large proportions of missing data for father’s own education, a proxy for his socioeconomic background. We did, however, have information on marital status (being married or having a partner) at the time of childbirth and age at childbirth which may capture some social factors, and we therefore included these as proxy variables for father’s socioeconomic background. We also conducted separate analyses for the subpopulation where information on father’s education was available and got almost the same estimates and 95% CI when adjusting for father’s age and education as when adjusting for father’s age and marital status. Therefore, we argue that using age and marital status is a satisfactory way to deal with missing information on father’s education.”

4. Incorporate visual aids, such as flowcharts or diagrams (like the directed 

acyclic graph mentioned), to illustrate data linkages and the analysis process. This can help in understanding complex methodologies.

Response: Thank you for pointing this out. Illustration of data linkages is added as Figure 1. See page 12, line 277 and 278

Results

The "Results" section presents a clear and comprehensive overview of the findings from the study. However, there are some areas for improvement to enhance clarity and readability.

1. The section lacks detailed explanations and interpretations of the observed differences in maternal characteristics. Provide more interpretation and discussion on the observed differences in maternal characteristics and their potential implications for the study's findings.

Response: According to our DAG (Figure 2), paternal factors associate with maternal factors that could be the mediators in examining the association between the exposure and outcome. We mentioned that maternal factors are relevant to our study in the method section (page 14, line 321-325): 

“When evaluating the association between family history of atherosclerotic disease and perinatal death, focusing on the total effect of paternal family history, paternal and maternal own risks factors such as hypertension, diabetes, total cholesterol, smoking, physical activity as well as infant factors such as birth weight, gestational age, SGA 2.5 percentile, foetal anomalies will be mediators (Figure 2), and should not be adjusted for.”

To address the reviewer’s concern on impact of maternal characteristics on finding of our study, we have now done additional analyses after excluding mothers with known risk factors for perinatal death. See Method section, page 15, line 326-330:

“Nonetheless, to differentiate the effect of paternal family history from maternal known risk factors for perinatal death, we did additional analyses where mothers with some important risk factors for perinatal death such as preeclampsia including HELLP syndrome and eclampsia, pregestational and gestational hypertension, and pregestational and gestational diabetes were excluded.” 

See result section, page 29, line 540-541: “The resulting point estimates and 95% CI were almost the same as in the total population”. The resulting point estimates and 95% CI were presented in Supporting tables: Supporting Information 1, Table S3 and S4. 

We have further elaborated on the description of maternal characteristics in the result section (page 26, line 470-490): 

“Family history of CHD

The mothers whose partners had family history of CHD were slightly younger when delivering (26.7 vs 26.9 years) and had a slightly higher prevalence of preeclampsia (2.9% vs 2.6%) compared to those whose partners did not have family history of CHD, but differences were very small. Only a small proportion of them had chronic hypertension (0.2%), pregestational (0.2%) and gestational diabetes (0.2%), preeclampsia including HELLP syndrome and eclampsia (2.7%). Again, the differences between those with and without family history were small. Parity and the proportions of placenta previa and abruptio placenta were the same between the two group of mothers. 

Family history of Stroke

Mothers whose partners had family history of stroke were slightly younger at the time of childbirth compared to their counterparts (27.3 vs 27.5 years). Parity, the proportions of mothers with chronic hypertension (0.3%), pre-gestational (0.3%) and gestational diabetes (0.4%), and pre-ecl

---

## [Editor Report · Decision Letter 1]

1 Nov 2024

Paternal family history of premature atherosclerotic disease and perinatal death: a population-based cohort study

PONE-D-24-10471R1

Dear Dr. Win,

We’re pleased to inform you that your manuscript has been judged scientifically suitable for publication and will be formally accepted for publication once it meets all outstanding technical requirements.

Kind regards,

Clive J. Petry, PhD

Academic Editor

PLOS ONE
---

## [Editor Report · Acceptance letter]

7 Nov 2024

PONE-D-24-10471R1 

PLOS ONE

Dear Dr. Win, 

I'm pleased to inform you that your manuscript has been deemed suitable for publication in PLOS ONE. Congratulations! Your manuscript is now being handed over to our production team.

Kind regards, 

on behalf of

Dr. Clive J. Petry 

Academic Editor

PLOS ONE